# A linear perception-action mapping accounts for response range-dependent biases in heading estimation from optic flow

Qi Sun[1,2,3‡]*, Ling-Hao Xu[4‡], Alan A. Stocker[5]*

**1** School of Psychology, Zhejiang Normal University, Jinhua, China, **2** Zhejiang Key Laboratory of Intelligent Education Technology and Application, Zhejiang Normal University, Jinhua, China, **3** Intelligent Laboratory of Zhejiang Province in Mental Health and Crisis Intervention for Children and Adolescents, Jinhua, China, **4** Dominick P. Purpura Department of Neuroscience, Albert Einstein College of Medicine, Bronx, New York, United States of America, **5** Department of Psychology, University of Pennsylvania, Philadelphia, Pennsylvania, United States of America,

‡ Co-first authors.
* sunqi_psy@zjnu.edu.cn, astocker@psych.upenn.edu

## Abstract

Accurate estimation of heading direction from optic flow is a crucial aspect of human spatial perception. Previous psychophysical studies have shown that humans are typically biased in their heading estimates, but the reported results are inconsistent. While some studies found that humans generally underestimate heading direction (center bias), others observed the opposite, an overestimation of heading direction (peripheral bias). We conducted three psychophysical experiments showing that these conflicting findings may not reflect inherent differences in heading perception but can be attributed to the different sizes of the response range that participants were allowed to utilize when reporting their estimates. Notably, we show that participants' heading estimates monotonically scale with the size of the response range, leading to underestimation for small and overestimation for large response ranges. Additionally, neither the speed profile of the optic flow pattern nor the response method (mouse vs. keyboard) significantly affected participants' estimates. Furthermore, we introduce a Bayesian heading estimation model that can quantitatively account for participants' heading reports. The model assumes efficient sensory encoding of heading direction according to a prior inferred from human heading discrimination data. In addition, the model assumes a response mapping that linearly scales the perceptual estimate with a scaling factor that monotonically depends on the size of the response range. This simple perception-action model accurately predicts participants' estimates both in terms of mean and variance across all experimental conditions. Our findings underscore that human heading perception follows efficient Bayesian inference; differences in participants reported estimates can be parsimoniously explained as differences in mapping percept to probe response.

**Data availability statement:** All data and modeling codes have been deposited on OSF: https://osf.io/t47v6/?view_only=659d-4bac8e4344f996eb8a51eb236d55.

**Funding:** This study was supported by National Natural Science Foundation of China (NSFC), China (No. 32200842) to QS and in part by the University of Pennsylvania (to AAS). The funders had no role in study design, data collection and analysis, decision to publish, or preparation of the manuscript.

**Competing interests:** The authors have declared that no competing interests exist.

## Author summary

Humans can estimate the direction of their self-motion (heading) from the associated visual motion pattern (optic flow) on their retinae. While these heading estimates are typically biased, previous studies have found quite conflicting bias patterns despite using very similar optic flow stimuli. Our findings demonstrate that these differences in participants' reported estimates can, to a significant degree, be attributed to differences in the response range within which participants were able to move their cursor to report their estimates. We introduce an efficient Bayesian observer model that provides a quantitative framework for analyzing these conflicting bias patterns in light of response range differences. The model assumes that perceived heading directions are identical for identical optical flow patterns, yet the reported heading directions are the result of an additional response mapping that linearly maps the percept to the reported estimate. This model fits the data well and suggests that participants' reported perceptual estimates are modulated by post-perceptual response transformations, in particular under conditions where they do not receive feedback. Our study is a reminder that psychophysical measurements necessarily provide only an indirect account of perception. Full explanations of such data require the inclusion of appropriate post-perceptual transformations that describe the mapping from perception to action.

## Introduction

An accurate and precise estimation of stimulus features based on visual information is important for a successful interaction with our environment. However, humans are often biased in their estimates of stimulus features. For instance, orientations are often inaccurately perceived, a phenomenon referred to as the oblique effect [1], while color estimates exhibit systematic compressions towards the center of individual color categories [2]. Most of these biases are robust and have been successfully replicated across multiple studies performed by different labs. For example, the effect that perceived visual orientations are typically biased away from cardinal orientations (the aforementioned oblique effect), is robust and has been repeatedly reported with little quantitative difference [3–5]. However, that is not true for biases in perceived heading direction (i.e., self-motion direction) based on optic flow (Fig 1A and S1 and S2 Movies). Some studies found that participants consistently overestimated their heading directions, thus exhibiting a peripheral bias [6–8]. Whereas in other studies, participants consistently underestimated their heading direction, producing a consistent center bias towards the straight-ahead direction [9–15]. What contributes to these seemingly contradictory results remains unclear, in particular since all these studies used very similar optic flow stimuli in overall similar experiments. A careful examination of the differences in experimental design between these studies, however, reveals two potential factors that may have caused the contrasting behavioral

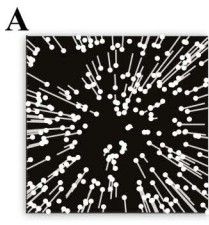
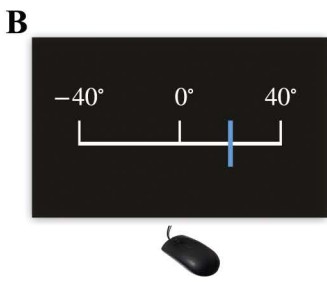
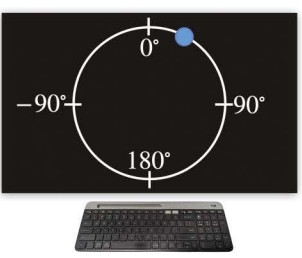
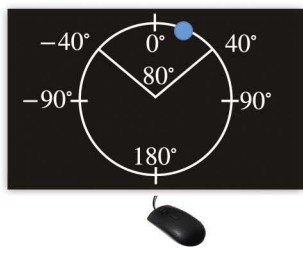

**Fig 1. Heading estimation based on optic flow.** (A) Illustration of an optic flow pattern [17] that simulates an observer moving straight-forward (i.e., a heading direction of 0°). The dots indicate the positions of dots in the first frame. The white lines (invisible in the experiment) indicate the motion trajectory of dots in the subsequent frames. Previous heading perception studies have used very similar optic flow stimuli. (B) Response methods used in our three experiments. In Experiment 1, participants either used a mouse-controlled probe on a horizontal line with a range of 80° (left panel) or moved a key-controlled dot on a circle with a range of 360° (middle panel) to indicate their perceived heading direction. In Experiment 2, all participants reported their perceived heading direction by moving a mouse-controlled probe on a line (block 1), circle (block 2), or an arc with a range of 80° (block 3, right panel). In Experiment 3, participants reported their estimates for three different arc ranges: 80°, 160°, or 240° (right panel). The range of tested heading directions in all experiments was [-33°, +33°]. Note, the white lines and the numbers denoting the range were not shown to participants during the experiment.

results. The first factor concerns the speed profiles of the used optic flow stimuli. Studies that found center biases (underestimation) generally utilized optic flow patterns that simulate observers moving in 3D space at constant speed [10–12,15]. In contrast, studies that reported a peripheral bias (overestimation) typically employed optic flow patterns that simulate observer moving at variable speeds [6–8,16]. Note that variable speed in this these studies refers to a predetermined pattern of acceleration and deceleration during stimulus presentation. The second factor is the response method. Specifically, in studies that reported center bias, participants expressed their perceived headings by adjusting the position of a probe along a horizontal line using the mouse (see right panel in Fig 1B; [10–12,15]), whereas in the studies that reported peripheral bias, participants adjusted the position of a probe on a circle using key presses (see middle panel in Fig 1B; [6,7]). It is unclear in what way these two factors are responsible for the different type of bias and why. S1 and S2 Movies provide examples of the optic flow stimuli used in the experiments, demonstrating constant and variable speed profiles, respectively.

In the current study, we conducted three experiments aimed at determining how stimulus speed profiles and response methods may affect heading estimates based on optic flow. Our findings reveal that the response range played a pivotal role in causing distinct estimation biases, causing a gradual transition from underestimation to overestimation biases with increasing response range. Building on these empirical results, we introduce a model that extends the efficient Bayesian observer model of Wei & Stocker [18–20]. The extended observer model assumes that perceived heading direction is linearly mapped to participants' reported estimates with a scaling factor that grows monotonically with the size of the response range. We found that our model accurately predicts both biases and variances in participants' heading estimates across different response ranges, suggesting that divergent heading estimates can be attributed to variations in response ranges. This novel extension of the efficient Bayesian observer model provides a potentially unifying perspective for the observed biases in heading estimation and highlights the importance of considering sensorimotor transformations when interpreting behavioral reports in perceptual decision-making tasks.

## Results

### Impact of navigation speed on reported heading direction (Experiment 1)

Thirty-six participants were exposed to two types of optic flow stimuli generated by simulated movement of observers in a 3D dot-cloud environment: movement was either at constant speed or at variable speed (S1 and S2 Movies). Throughout

all experiments, participants were exposed to heading directions within the range [-33°, 33°] with uniform sampling at 6° steps. Negative and positive values denote headings left (-) or right (+) of straight-ahead direction (0°). Consistent with previous studies, one group of participants (N = 18) reproduced their perceived heading directions by moving a mouse-controlled probe on a horizontal line with a limited response range (80°, left panel in Fig 1B), while another group adjusted a key-controlled dot on a circle (response range is 360°, middle panel in Fig 1B; see Methods for more details). No feedback was given to participants.

Fig 2 illustrates the bias (difference between the average heading estimate and the actual heading direction) and the standard deviation of participants' estimates for the two different speed and response conditions. The bias patterns are similar for the two speed profiles but exhibit significant differences under the two response conditions. Specifically,

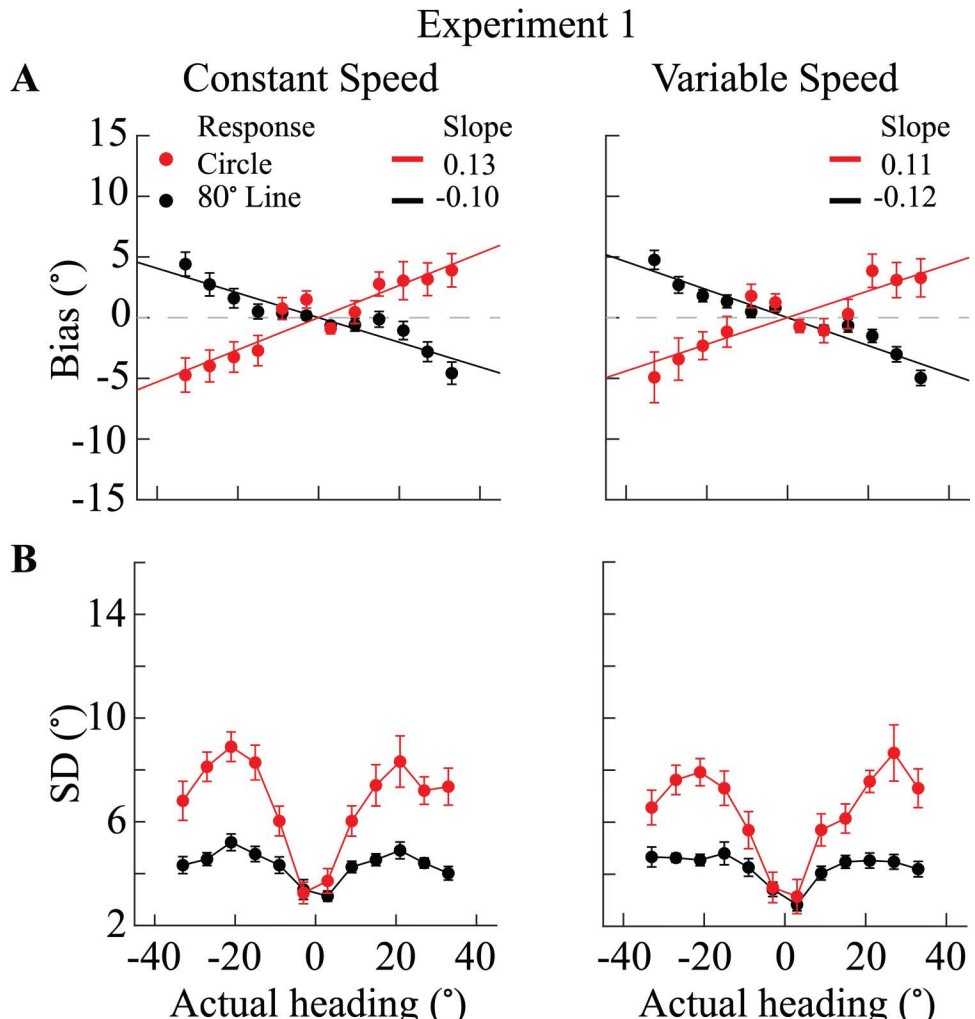

**Fig 2. Bias and standard deviation in perceived heading direction (Experiment 1, N = 18)** (A) Average bias across participants as function of actual heading angle (0° corresponds to straight-ahead). To capture the general tendency towards over- or underestimation, we applied a linear fit to the measured bias data points (solid lines). Positive slope values indicate overestimation (peripheral bias), while a negative slope indicates underestimation (center bias). (B) Average standard deviation (SD) of reported heading estimates across participants as a function of actual heading direction. Red dots correspond to estimates reported on the circle, and black dots correspond to estimates reported on the line. Error bars in all panels indicate the standard error across participants.

participants overestimated heading direction (peripheral bias) when reporting their estimates on the circle (red line), whereas they underestimated heading direction (center bias) when reporting on the line (black line). These results suggest that the speed profile by which participants navigate does not affect perceived heading directions. The specific response condition, however, can substantially affect the reported heading estimates.

Likewise, variability in participants' estimates is also differentunder the two response conditions but again does not depend on the speed profile. Standard deviations were significantly larger in the circle response condition compared to the line response condition, although the overall pattern was similar (Fig 2B).

## Impact of response method (mouse vs. keys) on reported heading direction (Experiment 2)

Experiment 2 was identical to Experiment 1, with the exceptions that i) the optic flow displays corresponded to an observer moving at a constant speed, and ii) participants reported their perceived heading direction by adjusting a cursor with a computer mouse. In addition to the line- and circle-response conditions (left and middle panels in Fig 1B), we also included an arc-response condition where participants reported their heading direction on an 80° response arc ([-40°, + 40°] range; right panel in Fig 1B). Note that the tested heading directions never exceeded a range of [-33°, 33°]. The three response conditions were randomly interleaved in the experiment (see Methods for details).

Fig 3A shows the average biases across all 18 participants in Experiment 2. Participants clearly underestimate heading directions when reporting their estimates both on the 80° line and the 80° arc. Note, the biases are not significantly different between the two conditions. In contrast, participants clearly overestimated heading directions in trials where they reported their estimates on the circle. Consistent with the data from Experiment 1, the standard deviation of participants' reported estimates was significantly larger in the circle response conditions compared to the line and arc response conditions (Fig 3C); again, there was no significant difference between the latter two conditions. Results of Experiment 2 indicate that the divergent bias patterns are solely due to a difference in response range (80° arc vs. full circle). The difference in response method had no effect on participants' reported heading direction.

## Impact of response range on reported heading direction (Experiment 3)

Results of Experiment 2 indicate that heading estimates may progressively increase when participants are provided with an increasingly larger response range. We recruited another 18 participants to conduct Experiment 3. The experimental design was similar to that of Experiment 2 except that participants only provided their estimates on a response arc with progressively increased response range (i.e., arcs of 80°, 160°, and 240°; see Methods for more details). Importantly, trials of the different range conditions were randomly interleaved. Fig 3B shows that participants' center bias (underestimation) is progressively reduced with increasing response range, eventually turning into a peripheral bias (overestimation) when the response range is 240° or more. Standard deviations of participants' reported estimates as a function of actual heading direction are similar in shape across all response conditions but progressively increase with increasing response range (Fig 3D). The results of Experiment 3 confirm that the response range is the main factor leading to different estimation biases and standard deviations.

To further validate the influence of response range on reported heading estimates, we conducted an additional experiment in which participants reported their perceived headings by adjusting the mouse-controlled probe on a line of different lengths (e.g., 80°, 111°, and 142°) while the other experimental design parameters were the same as in Experiment 3. The results (S1 Fig) show that shorter lines led to underestimation while longer lines caused more overestimation, mirroring our findings with the arc responses. This highlights the generality of the range effect in heading perception based on optic flow.

In summary, the results from the three experiments show that participants' heading estimates were neither significantly different between different speed profiles (constant vs. variable) nor did they depend on the response method (mouse vs keyboard). However, estimates were strongly dependent on the response range, showing progressively increasing biases from underestimation to overestimation with increasing size of the response range.

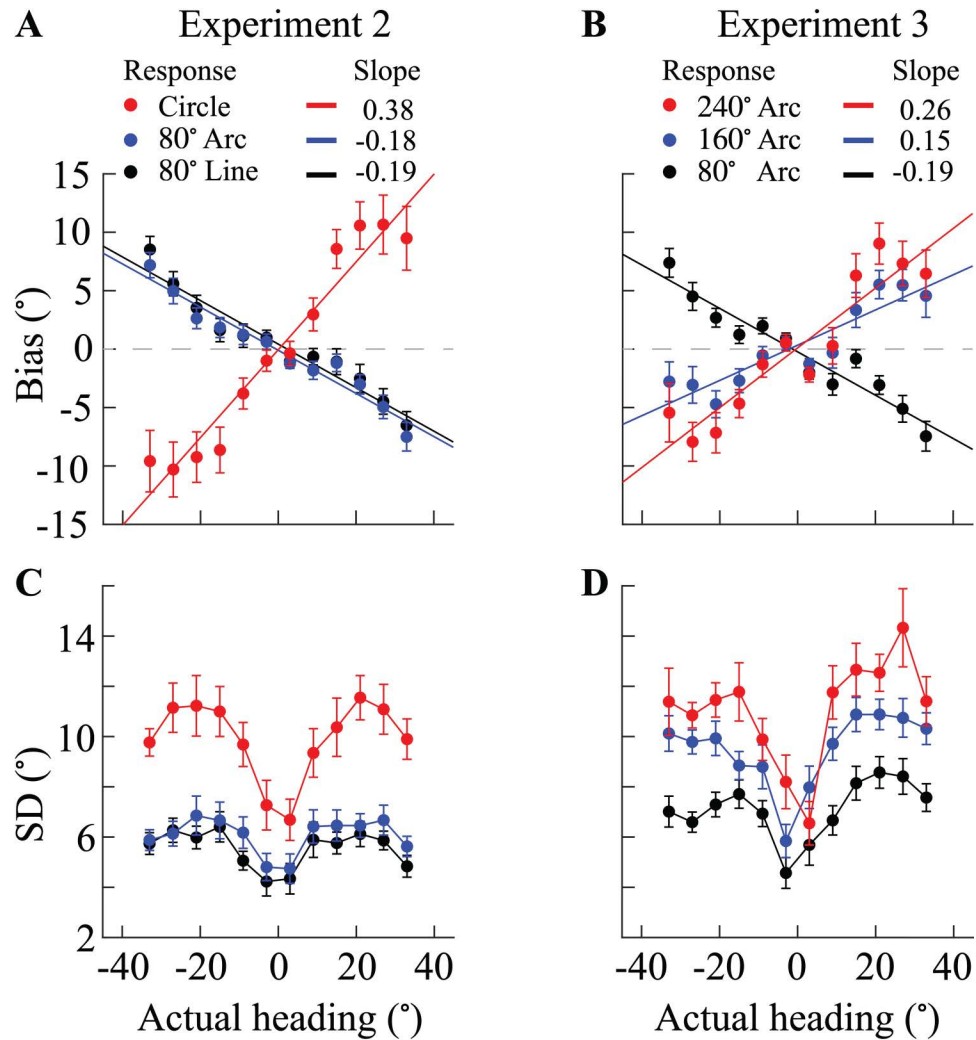

**Fig 3. Bias and standard deviation in perceived heading direction (Experiments 2 and 3, N = 18)** (A, B) Average bias of participants as a function of heading direction (0° corresponds to straight-ahead). Solid lines represent linear fits to the average bias curves. A positive slope indicates overestimation (peripheral bias), while a negative slope indicates underestimation (center bias). (C, D) Average standard deviation of reported heading estimates as a function of heading direction. In (A, C), the red, blue and black dots represent the circle, 80° arc, and line response conditions, respectively. In (B, D), the red, blue and black dots represent the 240°, 160°, and 80° arc response conditions, respectively. In all panels, error bars indicate the standard errors across participants.

## Bayesian observer model

Bayesian observer models express how knowledge about the distribution of a stimulus feature (i.e., the prior) needs to be combined with current sensory information (i.e., the likelihood) in order to optimally estimate the value of the feature [21–23]. Measured heading direction distributions of freely behaving human observers show a characteristic peak for straight-ahead heading [24]. With such a prior, however, standard Bayesian observer models consistently predict a bias toward the prior peak, thus a center bias toward straight-ahead direction [10–12,15]. As a result, these models cannot account for peripheral biases. More importantly, previous Bayesian observer model did not consider potential transformations between participants' heading percepts and their reported estimates, and thus are generally not equipped to explain the dependency on the response range we found in our experiments.

Here, we present a Bayesian observer model that can account for these range effects. The model extends the Bayesian observer model constrained by efficient coding proposed by Wei and Stocker [18–20]. The original model's innovation was to integrate efficient coding theory [25,26] with a Bayesian observer model, resulting in a unified framework known as the efficient Bayesian observer model. The model assumes that the neural system efficiently encodes the stimulus by maximizing the mutual information between the stimulus and its sensory representation. Subsequently, guided by the observer's prior belief p($\theta$), the efficient sensory representation is optimally decoded to yield the observer's estimate (Fig 3).

Originally, the model considers the Bayesian estimate as a direct reflection of participants' reported estimate (Fig 4A). With the extended model, however, we assume that participants' perceptual estimate of heading $\hat{\theta}$(m) is mapped to their reported estimate $\hat{\theta}_r$(m) via a linear response mapping, thus $\hat{\theta}_r$(m)=$\alpha_i\hat{\theta}$(m) (Fig 4B), with a scaling factor $\alpha_i$ that monotonically depends on the response range (see Methods).

We jointly fit this extended model to the data from individual participants in Experiment 3 across all three response conditions. We constrained the prior p($\theta$) on heading direction based on the assumption that heading direction is efficiently represented in the brain. This assumption predicts that the prior is inversely proportional to the discrimination threshold

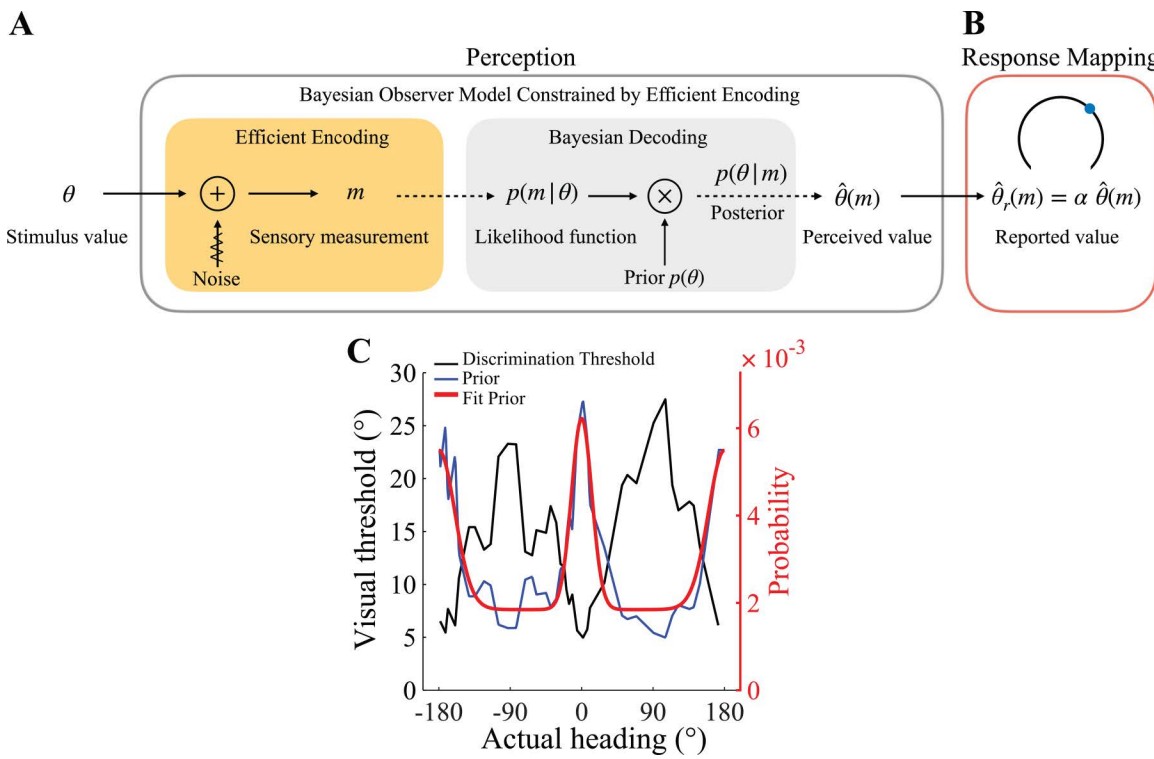

**Fig 4. Extended efficient Bayesian observer model.** (A) The efficient Bayesian observer model [19] assumes that the stimulus feature $\theta$ (e.g., heading direction) is first efficiently encoded in a sensory signal m before a Bayesian inference process computes an optimal estimate of the feature value $\hat{\theta}$(m). (B) We extended the efficient Bayesian observer model by assuming a response mapping that maps the perceptual estimate $\hat{\theta}$(m) to the participants reports $\hat{\theta}_r$(m) in the experiment. We assume that this mapping is linear, thus $\hat{\theta}_r$(m)=$\alpha_i\hat{\theta}$(m) where $\alpha_i$ is positive and monotonically depends on the size of the response range. (C) Prior distribution for heading direction, p($\theta$). We estimate a single prior for all participants based on previously measured heading discrimination thresholds D($\theta$) [8]. Under the efficient coding assumption, the prior is inversely proportional to D($\theta$) [20,28–30]. We smoothly approximate the experimentally determined prior with a two-peak von-Mises distribution, providing an indirect estimate of the underlying natural distribution of heading directions. S3 Fig shows an alternative method of obtaining a similar estimate of the heading prior based on neural response characteristics.

[20]. Specifically, we used a smooth approximation of previously measured heading direction discrimination data [8] to determine the prior on heading direction (see also [27]). The prior shows characteristic peaks for straight-ahead and straight-back directions (Fig 4C), in agreement with measured behavioral statistics [24]. Importantly, in the interest of simplicity, we used this fixed prior for all model fits of individual participants' data. Furthermore, given the deterministic generation of the optic flow stimuli, we assume no external noise. This leaves us with only two free model parameters per participant: the overall sensory noise magnitude $\kappa$ fixed across all response conditions, and the scaling factor $\alpha$ that is free for each response condition.

Note that the encoding precision, and thus the likelihood function, changes with heading direction but is precisely determined by the prior distribution (Fig 4C) according to efficient coding. Because we assume the same approximation of the prior distribution for all participants, the likelihood functions of each participant are self-similar and only scaled by the individual sensory noise magnitude (see Methods).

Fig 5 presents the results of our model fits. Fig 5A shows the fit scaling factor $\alpha$ (group level, individual subjects, and average), which exhibits a positive correlation with the expansion of the response range. Fig 5B shows the fit overall sensory noise level κ. Based on these two free parameters, our model generates detailed predictions for the bias and standard deviation of heading estimates, both at the group level (left panel of Fig 5C) and individual level (right panel of Fig 5C). These predictions demonstrate the model's ability to accurately represent the key characteristics of participants' data, both at the level of the average subject (Fig 5D) as well as at the level of individual participants (Fig 6). The model effectively anticipates biases (underestimation) when the response range is narrowest (80°). These biases transition to peripheral biases (overestimation) for broader response ranges (160° and 240°), with a discernible increase in bias corresponding to the enlargement of the response range. Similarly, the model adeptly mirrors the overall characteristic of the standard deviation as a function of heading direction, showcasing a gradual magnitude increase with increasing response range while its pattern remains the same (Fig 5E). This matches the observed self-similar pattern in participants' standard deviation (Figs 5F and 6).

In conclusion, our results show that an extended efficient Bayesian observer model based on a prior inferred from heading discrimination data can well account for nonlinear patterns of participants' heading estimation behavior (bias and standard deviation). Furthermore, the model assumes that participants' perceived heading direction is mapped to their reported heading direction with a scaling factor that monotonically depends on the specific response range available to the participants in the experiment.

It is important to note that for the standard efficient Bayesian observer model [19] the experimentally imposed response range is irrelevant; it would predict identical perceptual biases regardless of how responses are collected. Our extended model builds on this well-proven theoretical framework and thus assumes that the underlying perceptual (Bayesian) processing remains constant across conditions. With the proposed linear mapping between participants' percepts and their responses, it provides a parsimonious explanation for the observed systematic dependencies of participants' heading reports on response range without the need for unrealistic assumptions of how the (invisible) response range could possibly directly affect the percept. As such, our results provide an intuitive, non-perceptual explanation for the apparently contrasting findings of both center (underestimation) and peripheral (overestimation) biases in heading estimation from optic flow.

## Discussion

Our study demonstrates that the seemingly conflicting reports about biases in human heading estimates based on optic flow may be attributed to the influence of the response range in the psychophysical experiments. Specifically, participants' reported heading directions transitioned progressively from underestimation to overestimation with increasing response range. The fact that both bias and standard deviation in participants' reported heading estimates systematically vary with response range, despite identical visual stimuli, suggests that the transformation between perception and response plays

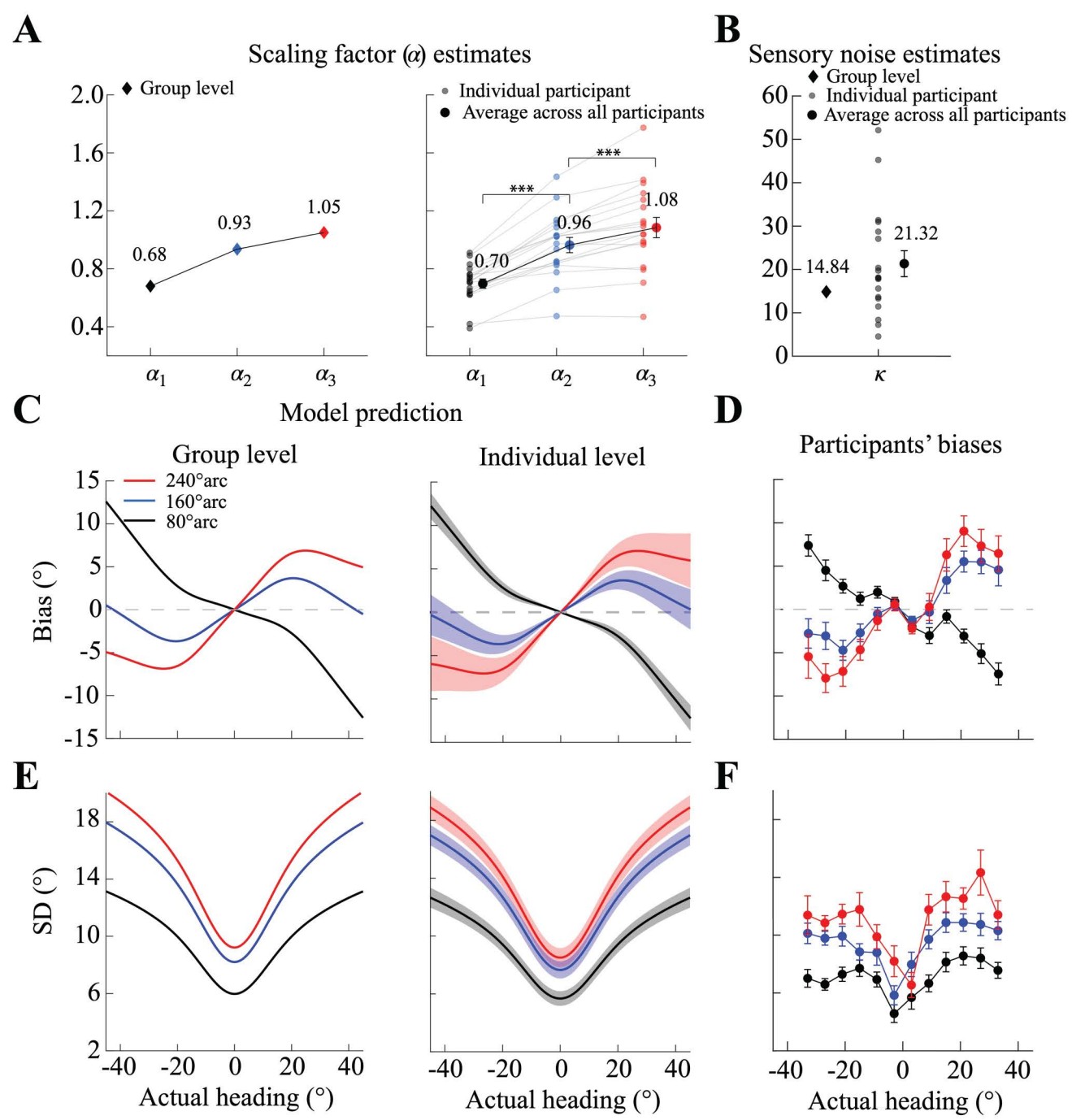

**Fig 5. Fit of the extended efficient Bayesian observer model to the data from Experiment 3.** (A) Values of the scaling factor $\alpha$ from the group-level fit (left panel; combined data from all participants) and individual-level fits (right panel; fits to each participant's data). The right panel also shows the mean $\alpha$ values across participants for each response range condition. A repeated-measures ANOVA with Bonferroni-corrected post-hoc tests revealed that $\alpha$ significantly increases with the response range (***$p < 0.001$). (B) Fit values of the sensory noise parameter $\kappa$ (group-level and individual-level fits). (C, E) Model predictions for bias and standard deviation of heading estimates, respectively, based on the group-level (left panel) and individual-level (right panel) fits. Shaded areas represent the standard error across the 18 participants. (D, F) Mean bias and standard deviation of heading estimates across the 18 participants in Experiment 3. Error bars indicate the standard error of the mean across participants.

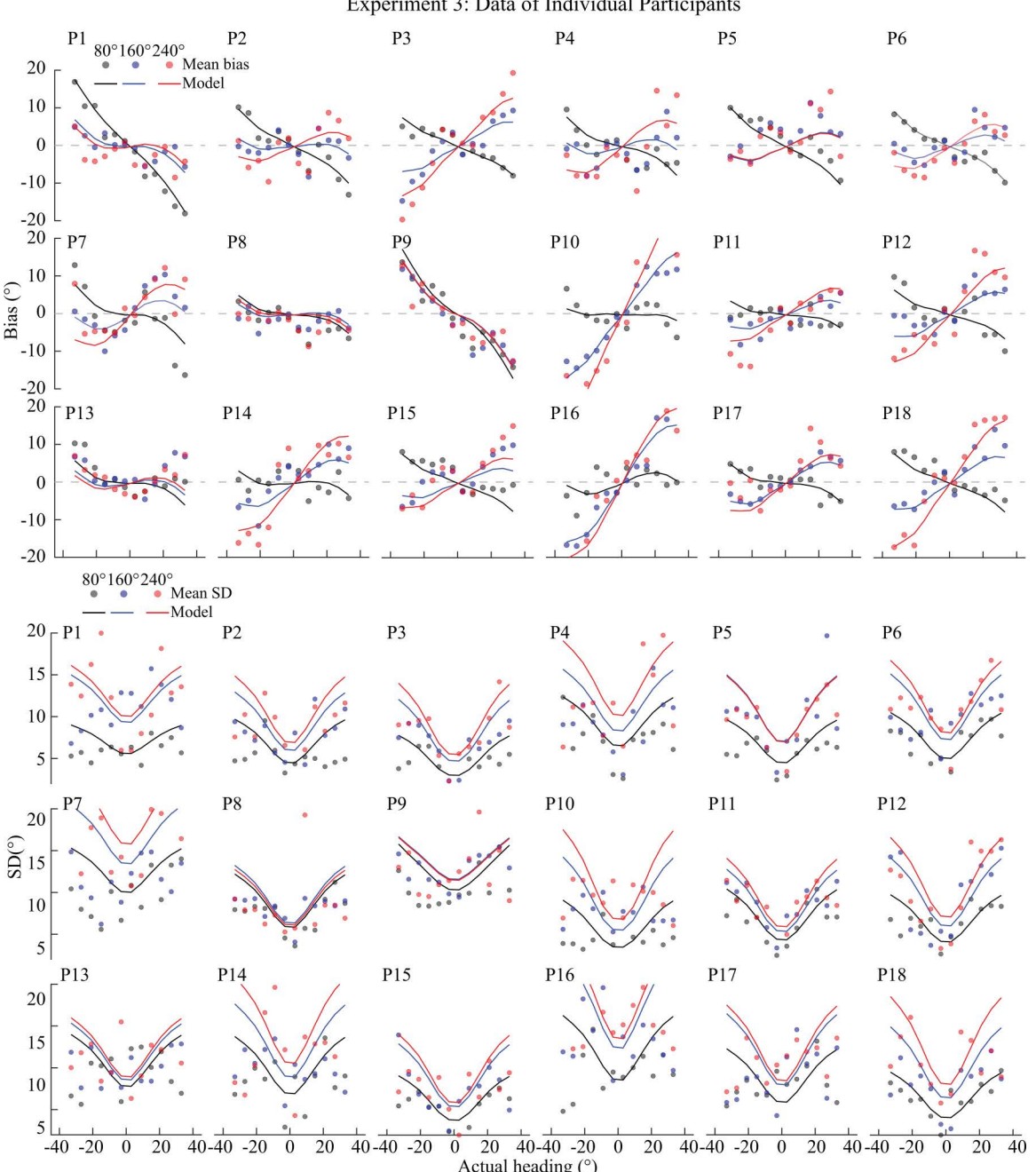

**Fig 6. Behavioral data and model fits for individual participants in Experiment 3, demonstrating the model's ability to capture individual-level patterns in heading estimation biases and variability.**

a crucial role in shaping participants' reported estimates. We propose that this systematic dependency is parsimoniously explained by extending the efficient Bayesian observer model [19] with a linear response mapping between perceived and reported estimates that scales with the response range (Fig 5A). We quantitatively validated this extended model and show that it provides a rational explanation for human heading perception, thereby potentially reconciling the many,

seemingly conflicting, previous findings about the nature of heading estimation biases. Our results emphasize the significant role that perception-action mapping can have in shaping the reported estimates of stimulus features in psychophysical experiments. The role has been frequently neglected in the past.

With only a few free parameters, our observer model is able to accurately express the large range of individual differences in participants' heading estimates, both in terms of bias and standard deviation (Fig 6). For our model, we assumed a fixed prior for heading direction based on the smooth approximation of previously recorded heading discrimination data [8]. According to the efficient coding assumption of the model, the prior is defined as being proportional to the inverse of the discrimination threshold function [20]. Although this is an indirect way of determining the prior, previous studies have shown that it faithfully captures the statistical prior in the visual environment for other stimulus variables such as local orientation [28] and visual motion [29]. Furthermore, preliminary data on the actual heading direction statistics of freely behaving human observers seem to support the predominance of straight-ahead heading [24]. We also considered an alternative derivation of the prior based on neural responses. Specifically, we analyzed single cell recorded, neural activity in area MSTd of non-human primates in response to optical flow stimuli indicating heading direction [8]. Following a previous approach [19], we extracted Fisher information from the neural population responses in order to derive the prior distribution (see S3 Fig). This neurally derived prior for heading direction is similar to the prior from the psychophysical discrimination data and, importantly, leads to model predictions that are qualitatively well aligned with the current results [31]. This alternative derivation of the prior further validated the assumptions of our model and our general conclusions. Future studies, however, are necessary to determine in more detail the exact shape of the prior distribution by actual measurements of the natural statistics of human heading behavior.

Such statistical measurements of the natural prior of human heading directions will also help to explain another interesting characteristic of our data. Although our observer model overall well explains the data, there are slight but systematic deviations between data and model predictions, especially with regard to the bias pattern around straight-ahead heading (0 degrees). Under all response conditions, participants' bias curves are non-monotonic (i.e., they show a "wiggle") in the range of 0–15 degree heading direction, which is not accounted for by the model using the current prior. However, a prior with non-monotonic fall-off (i.e., with some side peaks close to straight-ahead heading) likely would capture this behavior. An alternative explanation is that these non-monotonic bias patterns may be the signature of a categorical influence on participants heading estimates [32].

Our model makes clear predictions with regard to stimulus (external) and sensory (internal) noise (Fig 4A). Due to the efficient coding assumption, the two noise sources differentially affect the perceptual estimate [19]. In our experiments, the optic flow pattern was fully coherent with each simulated heading direction and thus the stimulus noise was zero. Furthermore, dot densities and contrasts, as well as stimulus presentation times were fixed, thus sensory noise was assumed to be constant. Our model predicts that adding stimulus noise (e.g., by reducing the fraction of dots with coherent optic flow motion) would lead to an increase in center bias (i.e., a decrease in peripheral bias, respectively), while an increase in sensory noise (e.g., by reducing the presentation time of the flow pattern) would decrease center bias (i.e., increase peripheral bias, respectively). Note, our model provides these quantitative predictions at the level of both, bias and standard deviation across the entire heading direction range. Future studies employing targeted psychophysical experiments will help to more specifically test those predictions.

It is important to note that in our experiments, participants did not receive visual feedback about the correct heading direction after reporting their estimates. This was a deliberate choice in order to be consistent with the experimental designs of previous studies. Without feedback, it seems reasonable to assume that participants used their experience of the response range to calibrate their responses. This would explain the monotonic scaling of the estimates with the size of the response range that we found in our study. Several studies have shown that real-time feedback can improve visual perception [5,33–35]. Thus, the degree to which feedback reduces of even eliminates the effects of the response range on heading estimation remains an interesting open question. Based on our model, we predict that with feedback, participants

will be able to quickly recalibrate and substantially reduce the observed bias because the biases are mostly attributed to the linear response mapping. We expect that the remaining biases will then mainly reflect only the inductive biases that are intrinsic to the Bayesian inference process due to the prior and the non-homogeneous, efficient encoding.

Another important aspect of our experimental design is that we manipulated the response range while keeping the stimulus range constant. Notably, in studies demonstrating conflicting bias patterns [6–15], researchers employed matched stimulus and response ranges (i.e., large stimulus range with large response range, and vice versa). Thus it is possible that the observed bias patterns may result from the coupled variation of both ranges rather than the response range alone. However, results of a recent study show that when stimulus range was systematically varied while maintaining response range constant, participants' bias patterns did not change significantly [36]. Accordingly, our current study focused on manipulations of the response range while maintaining stimulus range constant. Nonetheless, future studies are necessary to more systematically explore whether and how our results generalize to larger ranges of heading directions. This will also reveal whether the linear response mapping of our model extends uniformly across all heading directions or shows deviations at ecologically significant angles (such as straight ahead, sideways, or backward motion). This consideration of broader heading ranges also constrains our current version of the model. While we considered the scaling factor $\alpha$ a free model parameter, we note that for cases where both stimulus and response ranges span 360°, $\alpha$ must be constrained to equal 1 to avoid impossible predictions at extreme angles. Interestingly, with our unconstrained model fit, $\alpha$ is indeed approximately 1 for the largest response range. Thus, imposing such a constraint would not significantly affect our results.

Similarly, although our current experiments focused on forward heading directions only, our model's predictions for backward heading perception are well-supported by existing literature [6,7,37]. This suggests that our efficient Bayesian observer model and the assumed heading prior likely generalizes across the full range of heading directions. However, future research is needed to further validate our model for backward heading perception using similar experimental paradigms as those employed in our current study.

Finally, while our results well support the idea of a linear mapping between perceived and reported heading estimates, we acknowledge that this process likely involves working memory and other downstream processes. Thus, our results do not make any specific predictions about the underlying neural mechanisms involved in this perception-to-response mapping.

In conclusion, our study reveals that previously reported, contrasting findings of underestimation [9–15] and overestimation of heading direction [6–8] may be reduced to the influence of the different response ranges used in these different studies. We show that participants' reported estimates under different response range conditions are quantitatively accounted for by an extended efficient Bayesian observer model that incorporates a linear mapping between perceived and reported estimates. This suggests that human heading perception follows efficient Bayesian inference and is not dependent on the specific response characteristics of a psychophysical experiment; the observed differences in participants' reported estimates originate from differences in mapping from percept to probe response. As such, our model proposes a potential unifying account of a wide range of heading estimation datasets. It offers a simple yet parsimonious explanation for a seemingly complex phenomenon. Our findings contribute to a more comprehensive understanding of human perception and have implications for the design of navigational aids and virtual reality systems.

## Materials and methods

### Experimental methods

**Ethics statement.** The experiment was approved by the Scientific and Ethical Review Committee in the School of Psychology of Zhejiang Normal University. We obtained the consent forms of all participants before they conducted the experiments.

**Participants.** All participants were enrolled in Zhejiang Normal University. All were naive to the experimental purpose with normal or corrected-to-normal vision. Thirty-six participants took part in Experiment 1 and were randomly divided

into two groups (Group 1: 11 females, 7 males, 19–28 years; Group 2: 11 females, 7 males, 20–25 years). Eighteen participants took part in Experiment 2 (10 females, 8 males; 19–27 years). Another eighteen participants took part in Experiment 3 (9 females, 9 males; 17–23 years).

**Apparatus.** The experiment was programmed in MATLAB using the Psychophysics Toolbox 3 and presented on a 27-inch Dell monitor (resolution: 2560 H × 1440 V pixels; refresh rate: 60 Hz) with NVIDIA GeForce GTX 1660Ti graphics card.

**Stimuli.** The optic flow stimuli (80° H × 80° V visual angle, Fig 1A) simulated observers translating in a 3D dot-cloud (depth range: 0.20 – 5 m; eye-height: 0.17 m) consisting of 126 dots (diameter: 0.2°). The simulated self-motion speed was either constant (i.e., 1.5 m/s, used in Experiments 1–3; see S1 Movie), or first accelerating (0 m/s to 3 m/s) and then decelerating to 0 m/s (used in Experiment 1; see S2 Movie). Acceleration and deceleration were identical in magnitude (12 m/s$^2$). Two types of speed profiles generated the same moving distances (0.75 m) during the stimulus presentation. The simulated self-motion direction (i.e., heading) was selected from ±3°, ±9°, ±15°, ±21°, ±27°, and ±33°. Negative and positive values denote the headings left (-) or right (+) to the straight-ahead direction (0°).

## Procedure

Participants sat in a dark room with their heads stabilized with a chin-rest. They viewed the display binocularly with a 20-cm viewing distance. During the experiment, participants were asked to fixate on the display center without head and body movements during the entire experiment duration. Straight-ahead direction was aligned with the display center.

The trial procedures were the same across all three experiments. Each trial started with a heading display for 500 ms, followed by participants' responses (self-timed). The next trial started right after a participant's response. Heading direction in each trial was randomly selected from ±3°, ±9°, ±15°, ±21°, ±27°, and ±33°.

**Experiment 1** used a mixed experimental design. Response methods (line vs. circle responses) were the between-subject variable, while speed profiles (constant vs. variable speeds) were the within-subject variable. Thus, the current experiment included four conditions. Note, in the line-response condition, the left and right end-points of the line were shifted from the display center (0°) by 40°, resulting in a response range of 80° (left panel in Fig 1A). Participants used the mouse to adjust the probe position on the line to indicate their heading estimates. In contrast, in the circle-response condition, participants could freely adjust the probe position on a circle, resulting in a response range of 360° (middle panel in Fig 1A). Participants reported their heading estimates by pressing the arrow keys to adjust the probe dot position on the circle. The two speed profiles were tested in two separate blocks. That is, in the line- or circle-response conditions, participants completed two blocks. Each block corresponded to one speed profile. In each block of the line- and circle-response conditions, every heading direction was repeated 40 and 15 times, leading to a total of 480 and 180 trials, respectively. There were fewer trials in the circle-response condition than in the line-response condition because participants typically took more time to respond in the circle condition. The sequence of the two blocks in the line- and circle-response conditions was counterbalanced among participants. For example, one participant would first finish the block with the constant speed, followed by the block with the variable speed; another participant would first finish the block with the variable speed, followed by the constant speed.

**Experiment 2** was similar to Experiment 1, except that **i)** the motion speed was always constant, **ii)** the probe on the line or circle was controlled only by the mouse, and **iii)** each participant in the current experiment finished three blocks of trials. Each block corresponded to one response method (e.g., line, arc, or circle) and consisted of 240 trials (20 trials × 12 headings). The response range for the arc condition was 80° (right panel in Fig 1A). The conducting sequence of the three blocks was counterbalanced among participants.

**Experiment 3** was similar to Experiment 2, except that only one block was conducted. The block contained 12 heading directions and three response range conditions: 80°, 160°, and 240°. There was a total of 36 combinations. Each combination was repeated 15 times, resulting in 720 trials. All trials were randomly interleaved.

Before starting the experiment, participants were given 20 practice trials in which the speed profile was constant. The practice trials helped participants to familiarize the stimuli and response methods. After the practice part, the experiment started.

## Modeling methods

Our model consists of two parts: the efficient Bayesian observer model as originally proposed (Fig 3A, [19,20]) and a linear perception-action mapping stage (Fig 4B).

### Part 1: the efficient Bayesian observer model

The efficient Bayesian observer model was built based on the following three assumptions:

(1) The neural system encodes stimulus features (e.g., heading direction from optic flow) efficiently, following the efficient coding hypothesis [21,22];

(2) The sensory noise follows a von-Mises distribution in sensory space. Because the noise is homogeneous in this space, the likelihood function $p(m|\theta)$ for a stimulus $\theta_0$ also follows a von-Mises distribution, where $\mu$ is the distribution mean equal to $F(\theta_0)$, and $\kappa$ is the concentration parameter that represents the sensory noise magnitude;

(3) Following assumption (1) and [20], the prior distribution $p(\theta)$ is inversely proportional to the discrimination threshold $D(\theta)$, thus

$$p(\theta) \propto \frac{1}{D(\theta)} \tag{1}$$

In the current study, we used the visual discrimination data from [8] and fitted a bimodal von-Mises distribution to obtain a smooth approximation of the prior distribution (Fig 4C). Alternatively, we extracted the prior distribution for neural data based on single-cell recordings in area MSTd of macaque monkeys [8]; the resulting estimate of the prior distribution is similar (see S4 Fig).

With the above assumptions, we provide the following step-by-step formulation of the extended efficient Bayesian observer model:

**Encoding and mapping function $F(\theta)$.** To encode the heading direction, we employed a one-to-one mapping to transform the heading direction $\theta$ from stimulus space into sensory space according to efficient coding, given by

$$F(\theta) = \int_{-\pi}^{\theta} p(\vartheta) d\vartheta \tag{2}$$

which represents the cumulative prior distribution $p(\theta)$ with $\theta$ in [-π, π]. The cumulative prior distribution serves as a mapping function that transforms the physical stimulus space into a perceptual space. In this transformed space, the Fisher information becomes uniform, which, under the specific assumptions of the model, provides an efficient coding of sensory information. This approach has been widely adopted in perceptual modeling studies (e.g., [5,18–20,28,33,38]).

**Sensory measurement $m$.** For each heading direction $\theta_0$, the corresponding measurement on the sensory space $m$ is given by

$$m = F(\theta_0) + \eta \tag{3}$$

where $\eta$ represents the internal sensory noise.

**Likelihood function** $p(m|\theta)$. We assume the sensory noise $\eta$ follows a von-Mises distribution in the sensory space:

$$p(m|\theta) = \frac{1}{2\pi I_0(\kappa)} e^{(\kappa \cos(m-F(\theta)))} \tag{4}$$

where $I_0(\cdot)$ is the modified Bessel function of order 0, and $\kappa$ is the concentration parameter that controls the noise magnitude. The likelihood function $p(m|\theta)$ captures the bottom-up sensory information for a specific measurement $m$.

**Prior distribution** $p(\theta)$. The prior distribution $p(\theta)$ represents the observer's belief about the probability of each heading direction $\theta$ before any sensory observation. As stated in assumption (3), we assume it to be inversely proportional to the heading discrimination threshold $D(\theta)$ measured in the experiment [8]. We keep this prior distribution fixed for all model fits and predictions of our model analysis.

**Posterior distribution** $p(\theta|m)$. With the likelihood function $p(m|\theta)$ and the prior distribution $p(\theta)$ defined above, the posterior distribution $p(\theta|m)$ can be computed using Bayes' rule:

$$p(\theta|m) \propto p(m|\theta)p(\theta) \tag{5}$$

**Bayesian estimate** $\hat{\theta}(m)$. To obtain the Bayesian estimate $\hat{\theta}(m)$ for a given measurement $m$, we minimize the expected loss based on a specific loss function $L(\hat{\theta}, \theta)$:

$$\hat{\theta}(m) = \underset{\hat{\theta}}{\text{argmin}} \int L(\hat{\theta}, \theta)p(\theta|m)d\theta \tag{6}$$

For the squared error loss function ($L_2$ norm: $L(\hat{\theta},\theta) = \left\|\hat{\theta}-\theta\right\|_2$, the Bayesian estimate is the mean of the posterior distribution:

$$\hat{\theta}(m) = \int \theta p(\theta|m)d\theta \tag{7}$$

**Estimation bias** $b(\theta_0)$ **and standard deviation** $\sigma(\theta_0)$. The expected bias $b(\theta_0)$ for a given stimulus value $\theta_0$ can be computed by marginalizing the estimate $\hat{\theta}(m)$ over the measurement distribution $p(m|\theta_0)$:

$$b(\theta_0) = \int \hat{\theta}(m)p(m|\theta_0)dm - \theta_0 \tag{8}$$

To compute the standard deviation of the circular variable $\hat{\theta}$, we first calculate the average cosine and sine of the estimate:

$$C(\theta_0) = \int \cos\left(\hat{\theta}(m)\right) p(m|\theta_0)dm$$
$$S(\theta_0) = \int \sin\left(\hat{\theta}(m)\right) p(m|\theta_0)dm \tag{9}$$

Then, the circular standard deviation $\sigma(\theta_0)$ of the estimate for a given stimulus value $\theta_0$ can be computed as:

$$\sigma(\theta_0) = \sqrt{2\left(1 - \sqrt{C(\theta_0)^2 + S(\theta_0)^2}\right)} \tag{10}$$

**Part 2: linear perception-action mapping**

To account for the effect of the response range on the reported heading estimates, we extend the efficient Bayesian observer model by introducing a linear mapping between the perceived heading direction $\hat{\theta}(m)$ and the reported heading estimate $\hat{\theta}_r(m)$:

$$\hat{\theta}_r(m) = \alpha_i \hat{\theta}(m) \tag{11}$$

where $\alpha_i$ (i=1, 2, 3) is a scaling factor that depends on the response range condition (Fig 5A).

**Part 3: model fitting**

Our extended model has four free parameters: the sensory noise magnitude $\kappa$ (assumed to be constant across response range conditions) and the three scaling factors $\{\alpha_1, \alpha_2, \alpha_3\}$ (one for each response range condition). We jointly fit these parameters to the data from all participants in Experiment 3 using maximum likelihood estimation:

$$\{\kappa^*, \alpha_1^*, \alpha_2^*, \alpha_3^*\} = \underset{\kappa, \alpha_1, \alpha_2, \alpha_3}{\operatorname{argmax}} p(X|\kappa, \alpha_1, \alpha_2, \alpha_3) \tag{12}$$

where $X$ represents the observed heading estimates across all participants and conditions.

## Supporting information

**S1 Movie. Optic flow with constant speed.**
(MP4)

**S2 Movie. Optic flow with variable speed.**
(MP4)

**S1 Fig. Experiment in which participants report their perceive headings on a line with limited ranges.**
(EPS)

**S2 Fig. Bias and standard deviation in perceived heading direction with line-responses (N = 18).** The lengths of the response line are randomly selected from 80°, 111°, and 142°. The stimulus and apparatus parameter, as well as the experimental procedures are similar to those in Experiment 3. (A) Average bias of participants as function of actual heading angle (0° corresponds to straight-ahead). To capture the general tendency towards over- estimation or under-estimation, we fit a linear function to the measured bias data points (solid lines). Positive slope values indicate overestimation (peripheral bias), while a negative slope indicates underestimation (center bias). (B) Average standard deviation (SD) of heading estimates across participants as a function of actual heading direction. Red dots represent estimates reported on the circle, and black dots represent estimates reported on the line. Error bars in all plots indicate the standard error across participants. The raw data can be also downloaded from the OSF link: https://osf.io/t47v6/?view_only=659d4bac8e4344f996eb8a51eb236d55.
(EPS)

**S3 Fig. Comparison between the psychophysical prior and the neurally extracted prior.**
(EPS)

**S4 Fig. Comparison between the psychophysical prior distribution derived from human heading discrimination data (blue) and the neurally extracted prior from primates MSTd recording data (red).** The psychophysical prior (blue) is identical to that shown in Fig 4C The neurally extracted prior (red) is constrained by the Fisher information $J(\theta)$ from previously measured neural population responses in macaque MSTd area [8], following the approach described in [19]. Under the efficient coding assumption, the prior is proportional to the square root of the Fisher information $\sqrt{J(\theta)}$.
(EPS)

## Acknowledgments

We thank Bao-Yin Zhang and Lin-Zhe Zhan for collecting the data and doing the pilot data analysis.

## Author contributions

**Conceptualization:** Qi Sun, Ling-Hao Xu, Alan A. Stocker.

**Data curation:** Qi Sun.

**Formal analysis:** Qi Sun, Ling-Hao Xu, Alan A. Stocker.

**Funding acquisition:** Qi Sun.

**Methodology:** Qi Sun, Ling-Hao Xu, Alan A. Stocker.

**Project administration:** Qi Sun.

**Resources:** Qi Sun.

**Supervision:** Alan A. Stocker.

**Validation:** Qi Sun.

**Visualization:** Qi Sun, Ling-Hao Xu.

**Writing – original draft:** Qi Sun, Ling-Hao Xu.

**Writing – review & editing:** Qi Sun, Ling-Hao Xu, Alan A. Stocker.

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
