## [Decision Letter · Decision Letter 0]

Dear Dr. Sun,

Thank you very much for submitting your manuscript "A linear sensorimotor transformation accounts for response range-dependent biases in human heading estimation" for consideration at PLOS Computational Biology.

As with all papers reviewed by the journal, your manuscript was reviewed by members of the editorial board and by three independent reviewers. The reviewers appreciated the well-conceived and well-conducted experiments which convincingly show that the range of possible responses influence people's reports of heading direction. However, the reviewers also highlighted some substantial weaknesses. In particular, the reviewers felt that the conclusions were overstated, and raised concerns about the nature of the prior in the model and whether the model could account for other existing observations. If you believe that these weaknesses can be adequately address, then we would like to invite the resubmission of a significantly-revised version that takes into account the reviewers' comments.

We cannot make any decision about publication until we have seen the revised manuscript and your response to the reviewers' comments. Your revised manuscript is also likely to be sent to reviewers for further evaluation.

Sincerely,

Adrian M Haith

Academic Editor

PLOS Computational Biology

Tobias Bollenbach

Section Editor

PLOS Computational Biology

Reviewer's Responses to Questions

**Comments to the Authors:**

Reviewer #1: The current study examined heading perceptual bias reported in conflict in previous studies. The authors found that the conflict may be due to different range of perceptual report. Specifically, the bias is lager/overestimation when the scaled report range is large, and vice versa for the smaller bias/underestimation. The current study also excludes the motor report type, identifying the perceptual source. The finding is interesting.

Major comment:

The result from the polar-report is robust and persuasive, yet if it is true, it may apply to the line-report. That is, when using different scaled range of the line, the over/under-estimation bias may also apply. It is worthwhile to test whether under longer lines, there would be overestimation and under shorter lines, there would be underestimation.

Minor comment:

1. Summary: "while these heading estimates are typically biased..." the sentence has grammar error.

2. Discussion: "estimates cam be solely..." typo error

Reviewer #2: This is an interesting study, in which the authors convincingly show that biases in heading perception (underestimation / overestimation in different studies) are dependent on the range of responses that participants were allowed to utilize when reporting their estimates.

Major comments

1. The interpretation/ message inferred by ‘motor’, using terms like “motor mapping”, “motor stage” etc. seems imprecise (in relation the meaning of ‘motor’) and presumptuous. The authors themselves show that the results of these biases are independent of the motor action (mouse or keyboard control). So, clearly it is not motor per se. So what exactly is this mapping between? In fact, perhaps perception itself is affected by the range of options provided? And there are other candidate functions between perception and motor control (working memory?). I don’t see any evidence in the paper that justifies the assumption that this is motor mapping.

2. The connection between the linear mapping offered here, and the model results (that show non-linear over/under-estimates) is not clear. How does linear mapping result in these non-linear results? I assume that this is probably due the prior – but this is not explained well enough. Especially when it comes the to the results/ figures.

3. Related to the above point - the deviations in the figures (that show under/over-estimation) are fit with linear functions, yet those deviations are not linear. Moreover, Fig. 2a – the four innermost headings (‘circle’ response) show the opposite effect to the four outermost headings on either side. This difference is seen for both constant speed and variable speed – so it seems robust (and a similar effect is seen in Fig 3b). This effect/difference is lost when fitting a straight line that is mostly constrained by the more peripheral headings. A linear fit also does not seem to work well for the ‘line’ response data. The use of a linear fit for behavior that systematically deviates from linear, seems questionable. Also in Fig 3a,b there are systematic differences vs. linear. How are these reconciled with a linear mapping model?

4. While the authors convincingly show that the range of possible responses is important, clearly there are also other factors at play for determining the bias (such as heading prior, as the authors show). So, statements like “entirely attributed” in the author’s summary, and “fully explained” in the Discussion etc. seem over the top.

Minor comments

1. “two optic flow displays” is unclear, e.g., do the authors mean screens or stimuli?

2. ‘variable speed’ could be better defined. Sounds like speed fluctuates randomly during the stimulus. Although I eventually found the definition in the Methods, this could be better (at least generally, even without the specific details) explained earlier on in the text.

3. 20-cm viewing distance seems quite close. With the need to focus/ fixate on the display center during the entire experiment, could this lead to fatigue (especially for those with corrected vision) or other issues?

4. Some more explanations at other points in the paper can help. E.g., why does a cumulative distribution of the prior (Eq. 2) transform the heading direction from stimulus space into sensory space? The connection between the formula and its purported function are not explained.

Reviewer #3: The experiments are well-motivated and -conducted, and they may provide a plausible explanation for previously reported differences in heading biases across studies. However, I disagree with the strong conclusion expressed by the authors that these differences can be “fully explained by the influence of response range”. The authors do not address the fact that response ranges varied in previous studies because the stimulus ranges varied; the response range is typically expanded in proportion to the stimulus range. The same is not true in the present study; the stimulus range is held constant while the response range is manipulated. In this case, a response range that exceeds the stimulus range likely leads subjects to expand the range of responses they provide, as observed. This would not necessarily be the case if the stimulus range was similarly expanded. It is not clear that the current results would generalize to those more common conditions.

Concerning the modeling, one prominent weakness is that the Bayesian prior is derived from psychophysical data and then this prior is used to predict the psychophysical data of the current study. Thus, the model in this case does not provide a strong test of the Bayesian model, i.e. that perception depends on the prior probability. It also means that the match between observed psychophysical performance and that predicted based on previous psychophysical performance is not surprising.

Another concern is the predictions of the model for backward heading directions. First, the prior derived from discrimination performance shows a peak for backward movements, but I am doubtful that straight backward heading occurs as frequently as this prior suggests. Second, the linear scaling of the response relative to the percept would predict that overestimation (or underestimation) will persist even for backward heading directions, but prior studies that have measured perception of backward heading do not observe this pattern.

Overall, the primary contribution of the work is the demonstration that response range impacts responses. The exploratory modeling is also interesting, but the only addition relative to previous models is the linear scaling of the response relative to the perceptual estimate. Most of the weaknesses outlined above are not satisfactorily addressed.

**Have the authors made all data and (if applicable) computational code underlying the findings in their manuscript fully available?**

Reviewer #1: Yes

Reviewer #2: None

Reviewer #3: Yes

PLOS authors have the option to publish the peer review history of their article (what does this mean? ). If published, this will include your full peer review and any attached files.

**Do you want your identity to be public for this peer review?** For information about this choice, including consent withdrawal, please see our Privacy Policy .

Reviewer #1: No

Reviewer #2: No

Reviewer #3: No
---

## [Decision Letter · Decision Letter 1]

PCOMPBIOL-D-24-00875R1

A linear sensorimotor transformation accounts for response range-dependent biases in human heading estimation

PLOS Computational Biology

Dear Dr. Sun,

Thank you for submitting your manuscript to PLOS Computational Biology, and for patience while this resubmission was re-reviewed and evaluated by the editorial team. Two of the reviewers were satisfied with the revisions to the paper, but with some minor additional comments. However, one reviewer still expressed substantial concerns about the paper: 1) The experiments and theory do not specifically address the impact on behavior when the stimulus range is varied in tandem with the response range (as is usually the case in experiments). 2) The prior doesn't directly reflect statistics of natural behavior, but is derived indirectly from discrimination tasks and is thus somewhat circular. 3) It's not totally clear that effects related to backwards heading are well explained by the model; the inferred prior may be non-ecological for backwards heading and it's unclear whether the model can predict estimation biases when participants are given a 360° response range.

I sympathize with the concerns of Reviewer 3. However, I believe there is also a lot of merit in the paper, both empirically, in eliminating alternative explanations for discrepancies across previous studies, and conceptually, in showing that a model that adds a simple linear scaling can alter the direction of biases, which does not seem obvious and so I wouldn't consider a trivial extension to existing models.

I think the reviewer's concerns can be adequately addressed as follows:

1) Address explicitly how the model predictions are affected when the response range imposed in the experiment is changed. As I understand the model, the experimentally imposed range of responses is irrelevant, and so you would predict no difference. It would be helpful to include a paragraph either in the results or discussion that addresses explicitly what the model predicts in this case.

2) Using an inferred prior is not unreasonable, but please check the paper carefully for how prior is described and possibly revise where necessary to clarify. For instance, the abstract suggests that the model assumes efficient coding "according to THE natural prior of freely behaving humans", which is misleading. It would be more accurate to say "according to A prior..."

3) Please respond to Reviewer 3's question about studies that have investigated 360° of heading stimuli.

Please submit your revised manuscript within 30 days Mar 15 2025 11:59PM. If you will need more time than this to complete your revisions, please reply to this message or contact the journal office at ploscompbiol@plos.org. Please include the following items when submitting your revised manuscript:

We look forward to receiving your revised manuscript.

Kind regards,

Adrian M Haith

Academic Editor

PLOS Computational Biology

Tobias Bollenbach

Section Editor

PLOS Computational Biology

**Reviewers' comments:**

Reviewer's Responses to Questions

Reviewer #1: My questions have been addressed well. I do not have further questions.

Reviewer #2: Overall the authors have updated the paper appropriately. Congrats on the well-done paper

I have one last minor comment:

While the authors made changes in response to my first comment (about ‘motor’ mapping terminology) – the ‘motor mapping’ assumption is still quite evident in the paper, for example, in Figure 4b (and the Bayesian model description in general). IMHO, equating ‘motor’ with ‘action’ or ‘response’ (or everything that is not perception) is inaccurate and lacks nuance. Hence, I would encourage the authors to still further improve this aspect in their paper.

Reviewer #3: I thank the authors for responding to my comments. Unfortunately, I still maintain that this is a rather modest and incremental advance in our understanding, specifically the unsurprising observation that observers expand their responses to fill the provided responses range. This alone is not an impactful finding. If the study was able conclusively explain previously observed differences in heading bias reported across studies, this would add to the impact. However, the failure to manipulate stimulus range in proportion to response range does not allow reaching this conclusion. The insistence of the authors that they have conclusively demonstrated this is a significant weakness.

In response to my three main criticisms, the authors have added to the text. While these additions acknowledge the criticisms, they do not alleviate them.

The first main criticism is covered above. To convincingly address this criticism, it would be necessary to collect additional data in which stimulus range is also varied. The addition of a few sentences acknowledging this weakness in the discussion is not sufficient.

The second main criticism is the circularity of the model; deriving the prior from psychophysical data, then using that prior to predict psychophysical data. The authors have added a prior derived from neural data, however, the fact that neural and psychophysical data agree in this way has been previously reported, and once again, the prior is not derived from behavioral heading measures.

The third main criticism is that the model requires a prior that is peaked for backward heading directions. Given that human locomotion is almost always in the forward direction, such a prior is unlikely. I am unconvinced by the authors’ rebuttal of this point.

A related observation that is dismissed by the authors is that a linear transformation of responses has difficulties for heading directions beyond +/- 90 degrees. At this point, a bias towards “overestimation” of forward heading angles will become a bias toward “underestimation” of backward heading angles. In other words, the linear mapping cannot account for biases observed in studies that have investigated 360 deg of heading stimuli. In these studies, there is front-back symmetrical bias towards lateral heading directions. As I understand it, a simple linear response mapping cannot explain this front-back symmetry.

**Have the authors made all data and (if applicable) computational code underlying the findings in their manuscript fully available?**

Reviewer #1: Yes

Reviewer #2: None

Reviewer #3: Yes

PLOS authors have the option to publish the peer review history of their article (what does this mean? ). If published, this will include your full peer review and any attached files.

**Do you want your identity to be public for this peer review?** For information about this choice, including consent withdrawal, please see our Privacy Policy .

Reviewer #1: No

Reviewer #2: No

Reviewer #3: No

**Figure resubmission:**
---

## [Decision Letter · Decision Letter 2]

PCOMPBIOL-D-24-00875R2

A linear perception-action mapping accounts for response range-dependent biases in heading estimation from optic flow

PLOS Computational Biology

Dear Dr. Sun,

Thank you for re-submitting your manuscript to PLOS Computational Biology. Though Reviewer #3 was satisfied with some of the changes, they do have some remaining concerns that should be addressed. Most importantly, the Reviewer points out that certain statements in the paper make somewhat exaggerated claims about the extent to which the paper demonstrates that the model has been shown to account for previous findings. I agree with the Reviewer that, although the phenomenon discovered in the experiments and the accompanying theory provide a promising explanation for previous findings, further work is no doubt necessary before this could be validated as a general theory of heading estimation biases, and some statements in the manuscript could be adjusted to reflect this and acknowledge that further work will be necessary to more rigorously validate this theory.

I therefore request that you revise the manuscript according to these concerns. Overall, I don't find the current paper too far from striking a reasonable tone in terms of what is claimed, but certain parts of the paper could certainly still be refined in line with Reviewer 3's concerns. Please also pay attention to Reviewer 3's other comments which could help further improve the clarity and presentation of the theory and how it relates to prior work.

Please submit your revised manuscript within 30 days Jun 20 2025 11:59PM. If you will need more time than this to complete your revisions, please reply to this message or contact the journal office at ploscompbiol@plos.org. Please include the following items when submitting your revised manuscript:

We look forward to receiving your revised manuscript.

Kind regards,

Adrian M Haith

Academic Editor

PLOS Computational Biology

Tobias Bollenbach

Section Editor

PLOS Computational Biology

**Reviewers' comments:**

Reviewer's Responses to Questions

Reviewer #3: I thank the authors for addressing my previous comments. I maintain that additional revisions are needed to bring the claims in line with what the work has actually demonstrated.

My first main criticism was that the manuscript falsely claims that the current findings and model account for previously observed discrepancies in the heading perception literature, namely that different patterns of bias have been reported across studies.

These claims can still be found in the abstract, author summary, and main manuscript. I assert that this is a false claim because the current study only manipulates response range (not stimulus range). In contrast, in all previous studies in the literature, when response range was varied, stimulus range was also varied.

In order to demonstrate that a simple linear scaling of the response can account for different bias patterns in the literature, it would be necessary to vary both stimulus and response range, as has been done in previous studies. In contrast, in the current study, stimulus range was held constant and only response range was varied. Therefore, it is misleading to claim that the current results and modeling can conclusively account for previously observed discrepancies in reported heading biases. In fact the authors do concede this point in their response to reviewers: “As such, of course, we do not strictly demonstrate that our findings and conclusions generalize beyond the experimental conditions tested.”

Given this concession, it would seem necessary to remove all claims in the manuscript that the current model and results can conclusively explain previously observed discrepancies in the literature. Instead, the authors introduce a novel manipulation (of response range), observe novel effects of the manipulation (heading biases), and conceive a model to explain these effects. It would therefore seem appropriate to edit the abstract, author summary, and manuscript to more accurately reflect this specific contribution without making broader, unsupported claims about how the work generalizes.

My second main criticism, namely that the prior is circular has been sufficiently acknowledged and addressed.

My third main criticism has also been addressed. However, some additional commentary on this point may be warranted. In an additional paragraph in the discussion, the authors acknowledge that 360 stimulus range with 360 response range dictates that the linear scaling factor must be set to a value of 1. What they do not explicitly acknowledge is that setting this parameter equal to 1 is the same as eliminating their novel response mapping stage and reverting to pre-existing modeling approaches. Extending this reasoning, and in the spirit of parsimony, one might argue that such 360 studies are therefore preferable because they allow models with fewer parameters and also allow assessing sensory-perceptual processes while reducing (or perhaps minimizing) the risk of introducing response-scaling biases.

**Have the authors made all data and (if applicable) computational code underlying the findings in their manuscript fully available?**

Reviewer #3: Yes

PLOS authors have the option to publish the peer review history of their article (what does this mean? ). If published, this will include your full peer review and any attached files.

**Do you want your identity to be public for this peer review?** For information about this choice, including consent withdrawal, please see our Privacy Policy .

Reviewer #3: No

**Figure resubmission:**
---

## [Editor Report · Decision Letter 3]

Dear Dr. Sun,

We are pleased to inform you that your manuscript 'A linear perception-action mapping accounts for response range-dependent biases in heading estimation from optic flow' has been provisionally accepted for publication in PLOS Computational Biology.

Best regards,

Adrian M Haith

Academic Editor

PLOS Computational Biology

Tobias Bollenbach

Section Editor

PLOS Computational Biology

---

## [Editor Report · Acceptance letter]

PCOMPBIOL-D-24-00875R3

A linear perception-action mapping accounts for response range-dependent biases in heading estimation from optic flow

Dear Dr Sun,

I am pleased to inform you that your manuscript has been formally accepted for publication in PLOS Computational Biology. Your manuscript is now with our production department and you will be notified of the publication date in due course.

With kind regards,

Anita Estes
